# Quality of Care for Hypertension in Primary Health Care in South Africa: Cross-Sectional Feasibility Study

**DOI:** 10.3390/healthcare13192398

**Published:** 2025-09-24

**Authors:** Enos Muisaphanda Rampamba, Stephen M. Campbell, Brian Godman, Johanna C. Meyer

**Affiliations:** 1Department of Public Health Pharmacy and Management, School of Pharmacy, Sefako Makgatho Health Sciences University, Garankuwa, Pretoria 0208, South Africa; stephen.campbell@manchester.ac.uk (S.M.C.); briangodman@outlook.com (B.G.); hannelie.meyer@smu.ac.za (J.C.M.); 2School of Health Sciences, University of Manchester, Manchester M13 9PL, UK; 3Department of Pharmacoepidemiology, Strathclyde Institute of Pharmacy and Biomedical Sciences, University of Strathclyde, Glasgow G4 0RE, UK; 4Centre for Neonatal and Paediatric Infection, Institute for Infection and Immunity, City St. George’s, University of London, London SW17 0RE, UK; 5South African Vaccination and Immunisation Centre, Sefako Makgatho Health Sciences University, Garankuwa, Pretoria 0208, South Africa

**Keywords:** hypertension, quality of care, quality indicators, primary health care, South Africa

## Abstract

**Introduction**: Little is known about the quality of care for patients with hypertension in primary health care (PHC) facilities in South Africa, where most people receive care. **Objectives:** To test 46 quality indicators, developed previously, to assess and improve care; to assess the indicators’ clinimetric properties; and to recommend improvement strategies. **Methods:** A descriptive cross-sectional clinical audit in a purposive sample of 12 South African PHC clinics involving a retrospective review of 295 patient medical records. **Results:** A total of 45 of the 46 indicators were tested in the main sample (n = 295), of which 9 indicators could not be applied. Of the 36 applicable indicators, 22 could be applied and measured for ≥75% of the sample, while 14 were applicable to ≤50% of the sample. Only five indicators showed a quality of care score for ≥75% of patients. Overall, 82% and 92% of the sample had their blood pressure (BP) recorded in the last 12 months or in the previous 5 years for those aged >40, respectively, and 53.2% had a controlled BP. In the last 12 months, 30% of patients had a cholesterol record, 30% had their BMI recorded, 17% had a hypertension review with a medical practitioner, and 12% had received lifestyle advice. Only 38% received all clinically indicated antihypertensive medicines at their last visit. **Conclusion:** There were gaps in the quality of care for patients with hypertension, demonstrating the need for greater adherence to evidence-based guidelines, better data quality, and the use of electronic health information systems. Twenty-two indicators are recommended to address these gaps and improve the quality of care, patient outcomes, and the health care system.

## 1. Introduction

Non-communicable diseases (NCDs), including cardiovascular diseases, are becoming the principal causes of mortality in sub-Saharan Africa [1]. The greatest increase in NCD deaths globally is likely to be in Africa unless urgent action is undertaken [2]. Strengthening primary health care (PHC) across Africa is needed to respond to the growing prevalence of NCDs across the continent [3,4]. This includes developing and sustaining foundational elements critical to delivering quality health care services, such as the effective use of health information systems, excellence across all health care facilities, the safe and effective use of medicines, and sustainable financing mechanisms that support continuous quality improvement. This is crucial for South Africa’s success in achieving universal health coverage through its National Health Insurance initiative [5,6]. Whilst South Africa has evidence-based guidelines for managing patients with hypertension [7], alongside a National Strategic Plan to improve the management of patients with NCDs [8], there is currently an ineffective use of health information systems and no agreed-upon indicators or metrics to measure, and hence improve, the quality of care for these patients at the PHC level in the country [8,9,10,11,12,13]. The government of South Africa has since committed to improving routine data collection and reporting as part of strengthening processes and monitoring and evaluating in the management of NCDs, including hypertension [8], to help reduce the burden.

There have been concerns about the quality of care and control of blood pressure (BP) amongst patients diagnosed with, and managed for, hypertension at the PHC level in South Africa as well as across low- and middle-income countries (LMICs), given growing prevalence rates [3,14,15,16,17,18,19]. A quality indicator framework in primary care can be the first step to improving the future quality of care in PHC clinics, as this is seen as cost-effective and efficient in reducing coronary heart diseases and strokes [14]. The implementation of quality indicators is vital to assess current rates of compliance to agreed guidelines, which can improve patient outcomes [20,21,22,23,24,25].

Forty-six quality indicators derived from national and international hypertension guidelines have been developed using the RAND/UCLA Appropriateness Method. This involved a panel of nine members assessing the current management of hypertension at the PHC level in South Africa [26,27]. The indicators were expressed as discrete individual patient-level indicators and aggregated as facility metrics [26]. Discrete individual patient-level indicators refer to measuring specified health characteristics in a specified patient at a specific time or over a specified period (e.g., blood pressure (BP)). Aggregated facility metrics refer to measurements (averages, medians, proportions) that provide a summary view of the observations of individuals in each observed group, e.g., the percentage of patients with their BP measured and recorded in their records in the past six months [28]. However, indicators developed using consensus techniques have face validity only and need to be tested for the attributes of good quality indicators, which include their acceptability, feasibility, reliability, and validity [21,29,30,31]. In addition, they must be tested for their clinimetric properties, which refer to the quality of the measurement tool and the quality of its performance [32,33]. Testing includes the routine availability of data to measure indicator numerators and denominators reliably [32,33] (Box 1).

Box 1Attributes and clinimetric properties of clinical indicators [30,32,34].**Applicability**: An indicator is relevant to the specific population or context being monitored, e.g., PHC.**Feasibility**: Data are available in current PHC practice systems, i.e., electronic and/or hand-written medical records.**Measurability**: Data required to calculate the numerator and denominator of an indicator are readily available and can be collected with sufficient accuracy and consistency, e.g., data are available for ≥75% of a sample, and data are missing in <25% of cases.**Potential room for improvement**: The sensitivity of an indicator to detect variability in the achievement of the quality indicator between patients’ records and over time.**Validity**: Based on evidence-based guidelines.

Consequently, the objectives of this study were to test the quality indicators previously developed for the management of hypertension in primary care in South Africa, test the clinimetric properties of the indicators, assess the quality of care, and recommend strategies for the quality improvement of future hypertension management in PHC in South Africa, given current concerns.

## 2. Materials and Methods

### 2.1. Setting and Study Design

A descriptive cross-sectional retrospective clinical audit was conducted among 12 PHC clinics, purposively selected to include all four municipalities in one of the rural districts in a South African province. Forty-six hypertension quality indicators, which were developed using the RAND/UCLA Appropriateness Method and previously published [26], were applied retrospectively to 295 records of patients with a confirmed diagnosis of hypertension. The aim was to test the indicators for their applicability in the management of hypertension in primary care in South Africa.

### 2.2. Data Sources, Sample Size, and Data Collection

Data were collected from 22 January 2024 to 22 April 2024. The data collection tool was developed based on the list of 46 previously developed hypertension quality indicators in line with the principles outlined in the previously developed and validated indicator framework and indicator testing protocol [30]. A pilot study was conducted with the data collection tools applied in a PHC clinic that was not part of the 12 clinics included in the study. As a result of pilot testing the clinical audit tool, a duplicate indicator was removed. In addition, two indicators were amended for easy interpretation. This resulted in a total of 45 hypertension quality indicators to be used in the main study.

The estimated sample size was from 360 to 420 patient files, depending on the availability and accessibility of patient files from each clinic. Patient records of patients aged ≥18 years with an already confirmed diagnosis of hypertension were randomly selected from 12 PHC clinics by including every third patient record. Patients <18 years were excluded, as the majority of children and paediatrics are managed by specialists at the hospital level. In some clinics, records of patients with hypertension were stored together with records of other patients without hypertension. In these clinics, if the third patient record was not a patient with hypertension, the next patient record was selected until a record of a patient with hypertension was found.

Patient data were manually collected from 336 selected patient records that were marked as records of patients with hypertension. Whilst all 336 patients were on treatment for hypertension, there was no confirmation of hypertension diagnosis in 41 (12%) patients records. These patients were subsequently excluded, yielding a final sample size of 295 patient records.

Data were collected by the main researcher from all 12 clinics using the developed data collection tool. Using the developed data collection tool, all 45 indicators were applied to all patients’ records that met the inclusion criteria to test their measurability and applicability and to assess any improvement needed. All patients ≥18 years with a confirmed diagnosis of hypertension who attended the clinics during the retrospective 12 months of the study period were included. Each indicator was tested within its target population, which is referred to as the denominator of the indicator, i.e., the number of patients who should receive guideline-recommended specific clinical care, for example, taking and recording serum potassium concentration within 6 months of treatment for patients on spironolactone. The indicator was then measured for its applicability by counting the number of patients who received the recommended care, which is referred to as the numerator of the indicator. The denominator and numerator of the indicator denote the population to which the indicator is applicable, and the population to whom the indicator was applied in terms of the care they received.

### 2.3. Data Management and Analysis

Data were consolidated from across all 12 clinics to show aggregate facility scores as a percentage. Data collected from patient medical records were entered and analysed using an MS Excel^®^ spreadsheet. Whilst indicators were expressed as discrete individual patient-level indicators and aggregated as facility metrics [26], individual patient-level data were also analysed as aggregated facility metrics. In terms of clinimetric properties (Box 1), data availability as a function of measurability was considered optimal when data were available to measure and apply the indicator denominator and numerator in 100% of relevant patient records and was considered applicable if relevant and measurable to ≥75% of the qualifying patients’ records [32]. A cut-off value of 75% was used to define acceptable practices to accommodate instances where the guidelines are not applicable, data are unreliable, or the structures in which services are provided are sub-optimal, in the context of this rural province of South Africa. For example, BMI would not be determined in patients with physical disability, such as in patients who are wheelchair-bound, and patients’ pulses would not be checked within six months in those who default and come back after a long period. Another example includes insufficient health care practitioners in a facility to undertake clinically indicated activities [35,36]. A cut-off value of 75% was used to reflect previous low level of guideline compliance [36] and the fact that a clinician may have to apply his/her discretion in terms of risks involved. For example, a nurse may skip documenting care provided to rush to a pregnant woman who has just arrived by ambulance [37].

Clinimetric properties were determined by calculating the number of patients’ records that were relevant to each discrete indicator (denominator) and the total number of patients’ records within the denominator group where there were data to show if care met the indicator criterion (the numerator) [38], which is referred to as content validity. Content validity indicates the degree to which the data collection tool represents the domain of measurement. For example, if the tool was able to identify patients on spironolactone, and those on spironolactone in whom the serum potassium concentration was measured within 6 months of treatment, content validity was achieved [39].

The quality of care was expressed as the percentage of patients’ records that met the indicator criterion (numerator) within the total indicators that met the definition of the indicator (denominator). The percentage was calculated by determining the total number of applicable cases for an indicator (the denominator) and the total number of patients within the denominator group that met the indicator (the numerator) [38]. Final indicators were all expressed as aggregate facility indicators as a percentage.

### 2.4. Patient and Public Involvement

Patients and the public were not involved in the design, conduct, reporting, or dissemination plans of this research.

## 3. Results

### 3.1. Sample Characteristics

Data from 295 patient records of people aged ≥18 years who visited the clinic for hypertension reviews were collected from 12 PHC clinics (Table 1).

### 3.2. Clinical Audit of Patient Medical Records

Data were aggregated to create denominators and numerators to produce clinic-level results for each indicator as a percentage (Table 2).

### 3.3. Clinimetric Properties of the Indicators and Data Quality

Forty-five indicators were included in the main sample of 295 patient records. Table 2 shows that while 36 of the 45 indicators had data available to be able to calculate the denominator in patients who met the definition of the indicators, only 30 indicators had sufficient data available to calculate both the denominator and numerator to determine the quality of the hypertension management in the sample (Table 2). Nine indicators did not have data available to calculate the denominator and the numerator in order to assess the ongoing quality of care for the management of patients with hypertension in primary care in this rural province (indicator numbers: 4, 8, 16–18, 34–35, 40, 44).

Table 2 shows that 22 of the 45 indicators tested applied to ≥75% of the sample. Fourteen indicators applied to ≤50% of the sample for their management of hypertension, and nine indicators (indicator numbers: 4, 8, 16–18, 34–35, 40, 44) could not be applied.

### 3.4. Adherence to South African Hypertension Management Guidelines

Thirty of the thirty-six indicators that applied to this study sample had data to measure the numerator (Table 2). Only five indicators showed a quality of care score for ≥75% of patients (Table 2). The majority of patients (82%) had their BP recorded in the last 12 months, rising to 95% for those aged >40 in the last 5 years (Table 2). However, only 53.2% of the sample had their BP controlled in the past 12 months (Table 1). Thirty percent of the sample had a BMI recorded, and 41% of patients had their finger-prick blood glucose recorded in the last 12 months. Thirty-eight percent of patients had received all clinically indicated antihypertensive medicines at their last visit (Table 2). However, only 17% of the sample had a hypertension review in the last 12 months.

## 4. Discussion

We believe this is the first study to use quality indicators in a clinical audit to assess the management of patients with hypertension within public sector PHC clinics in South Africa, which is a public health priority in the country. The study sample had more females than males, a common trend that is seen at PHC facilities in South Africa, because women use health facilities more frequently than men [40].

This study aimed to test quality indicators developed for the management of hypertension in primary care in South Africa, test the clinimetric properties of the indicators, and assess the quality of care and recommend strategies for the quality improvement of future hypertension management in PHC in South Africa.

### 4.1. Clinimetric Properties of the Indicators

Most of the indicators (36 of 45) were applicable to the study population, i.e., they were patients who met the definition or criteria of the indicator, e.g., the indicator regarding conducting tests for serum potassium concentrations in patients who are on spironolactone would apply to patients on spironolactone and consequently providing the denominator for the indicator (Table 2). The denominator refers to the total population to which the indicator is applicable.

Once the indicator is applicable to a specific study population, the next step is to apply that indicator to determine the quality of care that patients with hypertension are receiving at the PHC clinics, which is found by calculating the numerator of the indicator. This could also be referred to as the measurement of the indicator performance. The numerator refers to the study population to which the indicator was applicable and to whom the recommended care that the indicator is measuring was provided. Again, most of the indicators (30 of 36) that were applicable to the study population had sufficient data available to calculate both the denominator and numerator to determine the quality of the hypertension management in the sample (Table 2).

Although all 45 indicators were appropriate and feasible for measuring the quality of care for patients with hypertension at the PHC clinics in South Africa, in this audit, nine indicators were not measurable (see Box 1, Table 2). They lacked sufficient data to calculate both the denominator (population that met the definition of the indicator) and the numerator (population in which the recommended care that the specific indicator is measuring was provided) for managing patients with hypertension. Consequently, they were not applicable for this study sample and location [31]. The nine indicators in question apply to groups of patients for which the indicator denominator may be rarely triggered. For example, this concerns pregnant women with severe pre-eclampsia and imminent eclampsia, patients with chronic kidney disease, and those in step 7 of the algorithm of hypertension management. This is because there were no patient files sampled for these indicators. This may be due to a small number of patients falling into these categories or no patients falling into these categories in these facilities. Whilst applicable to less than 10 patients (denominator), there were no data to calculate the numerator for the indicators, such as with patients with serum potassium concentrations recorded in their medical records in the past 6 months, or 12 months for patients on spironolactone or with an eGFR <30 mL/min, and for patients with hypertension for 10 years or more, respectively. Alongside this, this was also true for patients who had their therapy escalated to the next level of the hypertension algorithm management. There was also no data to calculate the numerator for patients with hypertension and a BMI of ≥27.5 kg/m^2^ or ≥30 kg/m^2^ in the preceding 12 months who had been referred to a weight management programme within 90 days of the BMI being recorded. There were no records in these patients’ files about the tests or services they should have received as per the hypertension guidelines, signalling poor adherence to current guidelines, because if not documented, it means care was denied or care was not provided [37]. Largely, this signals poor data quality for some key aspects of evidence-based hypertensive care. Twenty-two of the forty-five indicators in the audit applied to ≥75% of the sample with data that could be collected with sufficient accuracy and consistency (Box 1, Table 2) [32]. Overall, more than half of the indicators (22 of 36) that were applicable to this population performed well as they could be applied to ≥75% of the population.

### 4.2. Adherence to Hypertension Guidelines

While most indicators could be applied in this study population, the adherence to hypertension guidelines was generally poor. Adherence to guidelines could only be confirmed in less than a quarter (5 of 36) of situations (Table 2). This included the measuring of patients’ BP, in which compliance was higher among patients aged >40 years (Table 2).

Only half of the patients had controlled BP in the past 12 months (Table 1), something which can also be related to a poor or lack of adherence to hypertension guidelines [41]. Again, only a quarter of the patients had a BMI recorded, while just above a quarter of patients had a finger-prick blood glucose test recorded in the last 12 months. Equally concerning is that only just under a quarter of the patients surveyed received all clinically indicated antihypertensive medicines at their last visit (Table 2). Our study concurs with previous studies in South Africa and Zimbabwe where patients did not receive all pertinent antihypertensive medication and BPs were not controlled [42,43]. The challenges of medicine availability on the day of the consultation to PHC urgently needs to be addressed to promote adherence to both treatment guidelines and hypertension medicines among both prescribers and patients. Consequently, it is vital to ensure that all patients have access to agreed-upon medicines when visiting PHC clinics in South Africa. This involves the better forecasting of the medicine use and availability in PHC, building on current initiatives [44].

Less than a quarter of the patients had a hypertension review in the last 12 months. This poor adherence to hypertension guidelines has been a constant concern in PHC clinics in South Africa and helps explain the poor patient outcomes [41]. The accurate measurement of the quality of care depends on the availability of reliable data. However, this study shows incomplete data, a concern that has always existed, especially in primary care in LMICs [45], and which hinders the ability to routinely measure agreed indicators on a regular basis.

In the context of measuring the quality of care for patients with hypertension in PHC clinics in South Africa, it is possible that some of the actions described in the indicators are not applicable because, for example, escalating therapy for patients in PHC clinics to the next level of the hypertension management algorithm is often undertaken at the hospital or by visiting doctors to PHC clinics in South Africa. Similarly, Table 2 shows that there were six indicators for which the denominator was indicated as applicable for ≤10 patients (indicator numbers: 3, 5, 6, 12, 39, and 41) for their management for hypertension, and nine indicators did not apply to the patients in this sample.

The audit results in Table 2 highlight significant gaps in adherence to hypertension management guidelines [7], which urgently need to be addressed to improve the future quality of care for patients with hypertension in South Africa [22,24,41]. For example, the adherence to non-pharmacological lifestyle treatments and, where indicated, the adherence to pharmacological treatments can improve patients’ quality of life and outcomes [46,47]. However, 75 patients with hypertension and a BMI of ≥27.5 kg/m^2^ or ≥30 kg/m^2^ in the preceding 12 months were not referred to a weight management programme within 90 days of the BMI being recorded. Alongside this, whilst both height and weight were recorded in 117 patients to calculate the BMI (Table 1), this was recorded in only 89 (30%) of the 295 patients’ records in the past 12 months (Table 2). Whilst all hypertension medicines were dispensed to 112 (38%) patients [7,23,41,48], there was no data on prescriptions issued in 174 (59%) patients’ records for their last visit. This indicates poor recording and poor data management, which may lead to poor or lacking evidence-based clinical and administrative decisions, as both of these need reliable information systems that reflect the actual clinical processes.

Urine protein by dipstick was recorded in only three patients’ records, and less than half of patients (120; 41%) had their finger-prick blood glucose recorded in their medical records in the past 12 months, despite people with hypertension being at greater risk of insulin resistance than normotensive individuals [49]. Moreover, despite hypertension being a risk factor for chronic kidney disease [50], only 24 (16%) of 153 patients with uncontrolled hypertension had their serum creatinine concentration and eGFR recorded in their medical records in the past 12 months. There was no record of why the remaining patients did not receive the prescribed care, which reflects unreliable data to inform decision-making and to meet the requirements of the numerators of the relevant indicators. This study found that only 17% of the sample had a hypertension review in the last 12 months. This is a concern as evidence suggests that annual reviews can identify and address NCD risk factors [7,51], with annual reviews advocated in the South African National user guide on the prevention and treatment of hypertension in adults at the PHC level [7]. These findings resonate with previous studies that have found concerns with adherence to guidelines in South Africa and other parts of Africa [31,41,52] and elsewhere, including other LMICs [53]. This is important since adhering to evidence-based clinical guidelines improves patient outcomes [36,37,41,42].

These results provide preliminary data for each indicator, making it easier to identify the changes and strategies needed for quality improvement to monitor and manage hypertension at the PHC level in South Africa. For example, there were gaps in the data collected and the interpretation for decision-making, e.g., cases where there was a record that patients received lifestyle advice during their consultation; however, there were no records of their actual lifestyle, including whether the patient is a smoker or not. Again, most of the patients were overweight or obese, and this can explain diabetes as the highest comorbidity among these patients (Table 1). This raises concerns about whether patient empowerment through health talks, education, and counselling by the nurses is patient-specific. Continued overweight or obesity among patients can also indicate their non-compliance with lifestyle modifications in cases where they are aware of these interventions. Whilst available data such as this adds to the nursing load of manual data collection, it does not provide any information to assist with evidence-based decisions to improve the future quality of care and patient outcomes. This also needs addressing going forward.

South Africa aims to achieve an 80% target goal of BP control by 2030 among patients on treatment at the PHC level [4,8]. This study highlights that there is a long way to go to meet this objective. Currently, only 53.2%% of patients have a controlled BP, and this objective will not be met in the future unless a number of quality improvement plans are in place. The strategic documents about NCDs and hypertension management guidelines in South Africa lay the foundation to improve the quality of care that patients with hypertension should receive at PHC clinics [7,8]; however, this cannot be achieved without metrics to measure their progress and direct interventions [9,10,11,12,54,55,56].

This study provides preliminary results on the quality of care that patients with hypertension are receiving at the PHC level, and the changes needed to start to continually measure the quality of care at the PHC clinics to meet current goals. However, these results should be interpreted in context with the WHO five critical foundational elements of a quality health care system. These include health care workers; health care clinics; medicines, devices, and other technologies; information systems; and financing [57]. This is because quality improvement is a comprehensive exercise that requires a focus on multiple issues simultaneously to improve care and address current barriers and shortcomings in the health care system infrastructure that inhibit optimal quality of care for patients with hypertension in PHC clinics. For example, educational interventions on hypertension guideline adherence, investment in electronic data systems that host clinical templates, alongside patient education interventions for healthy lifestyles are all urgently required if South Africa is to achieve its goals for the management of patients with hypertension. The results from this audit are similar to other studies that previously audited manual patient records [32,41,58]. These results also expose the general trends of the data about the quality of care not being fully captured and being difficult to analyse due to a lack of standardised metrics and coding and the ways of reporting among LMICs, which includes South Africa [4]. In addition, data in paper-based systems is usually never used to continually assess the quality of care due to challenges in accessing and analysing it during audits. This is because such approaches need time and organised archives [4]. Consequently, manual data capturing may become infrequent if data is not readily available and being used. Considering the overburdened PHC facilities with chronic patients and a shortage of nurses, manual patient records and documentation will add to their workload, which may contribute to poor adherence to quality of care for patients with hypertension. Nurses sometimes omit some appropriate and necessary activities needed to provide quality care to patients due to inadequate time, as a result of a shortage of staff and other factors [36].

A primary care measurement framework platform hosting clinical templates could be the basis to improve the completeness, quality, and use of existing, preferably digital, medical record data to improve the management of patients with hypertension in PHC clinics in South Africa [59]. Pre-populated templates on patient medical records have a list of clinical actions that should be offered to the patient with HCPs, having only to insert the figures or narration, including the current BP, weight, height, and BMI, as clinically appropriate.

We recommend a stepwise approach to implement the hypertension quality indicators among PHC clinics in South Africa, which starts with the 22 indicators applicable to ≥75% of the sample. These must be used alongside implementation strategies that include educational interventions on guideline adherence and data quality and consistency as part of improvement plans, as well as patient education interventions on healthy lifestyles to improve the future quality of care of patients with hypertension in South Africa. This, together with improved collaboration amongst clinicians, could provide an improved quality of care in totality based on the WHO building blocks of health systems, as improving the quality of health care needs a multi-disciplinary approach and activities [4,57,60]. This becomes even more important as the success of the National Health Insurance plan that South Africa is engaging in should be based on local health care needs and high-quality primary care services and should involve patients, families, and communities [5].

### 4.3. Strength and Limitations

Although testing of the developed hypertension quality indicators was conducted in a small purposive convenience sample, as a translational study, we believe the study provides preliminary data for each indicator. Testing quality indicators is imperative to strengthen health programmes and enhance the quality and efficiency of clinical services as part of key approaches to quality improvement [30,31,34]. The strength of this study includes the data collection by an experienced researcher, maintaining the consistency and testing of indicators retrospectively based on current guidelines. The aggregated data were used to minimise data variations across the clinics due to the inconsistent data availability per indicator. On the other hand, inadequate or poor-quality data at the pilot clinics necessitate a large study. However, this was feasibility research and was not intended to generalise the results but rather to provide insight into future research projects, policy directions, and quality improvement.

Previous studies found 14% to 56% omissions in the treatment adherence guidelines [36]. Clinical care may be unpredictable or not recorded, leading to prospectively incomplete clinical documentation, which would then be interpreted as the clinical care received by the patient [37]. It is crucial that any difference in data outcomes between facilities is a genuine difference in the quality of care, rather than a function of the data quality and availability. While a 100% achievement is the ideal target to promote optimal quality care, as prescribed in the guidelines, in line with previous studies, a lower documentation (numerator) of clinical activities was used for 75% to test the clinimetric properties and the quality of the measurement tool [32,33,37]. This study was conducted retrospectively, which may introduce bias when the study population is not randomly selected from the target population and when there is loss of information and an inability to control confounders. Another consequence of retrospective study designs is that the data were originally collected for other purposes, which may lead to a lack of relevant information [61]. The limitations of this retrospective design were reduced by pre-defining the objectives of this study, the inclusion criteria, and the exclusion criteria and by defining the outcomes that would be used for measurement in this study. Despite these limitations, this study provides data on potential quality indicators that could be introduced into PHC clinics in South Africa and other LMICs to improve the future care of patients with hypertension.

## 5. Conclusions

Currently, there are no published indicators for measuring the quality of care that patients with hypertension receive at PHC clinics in South Africa. This study tested hypertension quality indicators in line with the hypertension management guidelines and in the context of routine PHC/ambulatory care practice in South Africa. This study has identified 22 indicators that applied to, and could be measured for, ≥75% of the sample. The study findings highlighted significant gaps in the current quality of care in South Africa regarding compliance with hypertension management guidelines. We recommend the implementation of the indicators in a larger quality improvement project, whilst concurrently working towards improving the adherence to current hypertension guidelines and instigating electronic information technology systems to obtain better quality data collection in real time to improve the quality of care and patient outcomes.

## Figures and Tables

**Table 1 healthcare-13-02398-t001:** Characteristics of the patient sample.

Characteristics	n	Mean (SD)	Median (Q1–Q3)
Age (years)	287	64.2 (13.9)	64 (55.0–72.5)
Weight (kg)	216	77.6 (18.6)	75.4 (65.0–88.0)
Height (m)	118	1.6 (7.1)	1.6 (1.6–1.7)
Calculated BMI	117	30.3 (7.4)	29.4 (25.6–34.3)
		Number (%)	
Sex			
Female	295	227 (76.9)	
Male	295	67 (22.7)	
Not specified	295	1 (0.3)	
Calculated BMI available	295	117 (39.7)	
BMI classification			
Overweight (25–29.9 kg/m^2^)	117	35 (29.9)	
Obese (≥30 kg/m^2^)	117	54 (46.2)	
Healthy weight (18.6–24.9 kg/m^2^)	117	25 (21.4)	
Underweight (<18.5 kg/m^2^)	117	3 (2.6)	
BP controlled	252	134 (53.2)	
Comorbidities	295	153 (51.9)	
Types of comorbidities			
Diabetes	153	112 (73.2)	
HIV and AIDS	153	24 (15.7)	
Other conditions	153	17 (11.1)	

NB: AIDS, acquired immunodeficiency syndrome; BMI, body mass index; BP, blood pressure; HIV, human immunodeficiency virus; Q1–Q3, quartile 1–quartile 3; SD, standard deviation.

**Table 2 healthcare-13-02398-t002:** Overall clinical audit results from 295 patient medical records from 12 clinics.

No.	Indicator	Total (Denominator)	Indicator Met (Numerator)	Overall Quality of Care (% of Patients)
1.	Percentage of patients in the practice/unit/facility with a BP recorded in the last 12 months	295	241	82%
2.	Patient with BMI recorded in the past 12 months	295	89	30%
3.	Patient had serum potassium concentration recorded in the past 6 months for patients on spironolactone or eGFR < 30 mL/min	7	0	0%
4.	Patient had serum creatinine concentration and eGFR recorded in the past 12 months for patients with proteinuria of 1+ or more	0	N/A	N/A
5.	Patient had serum creatinine concentration and eGFR recorded in the past 12 months for patients with existing cardiovascular disease	4	2	50%
6.	Patient had serum creatinine concentration and eGFR recorded in the past 12 months for patients with hypertension for 10 years or more	8	0	0%
7.	Patient had serum creatinine concentration and eGFR recorded in the past 12 months for patients with uncontrolled hypertension	153	24	16%
8.	Patient had serum creatinine concentration and eGFR recorded in the past 12 months for patients with chronic kidney disease (eGFR < 60 mL/min)	0	N/A	N/A
9.	Patients had finger-prick blood glucose recorded in their medical record in the past 12 months	295	120	41%
10.	Patient has had urine protein by dipstick in the past 12 months	295	3	1%
11.	Patients in the practice/unit/facility were screened for cardiovascular disease risk factors in the last 12 months	295	132	45%
12.	Patient in the practice/unit/facility checked for medicines and lifestyle modification adherence before escalating therapy	4	0	0%
13.	Patient aged 40 years and over with a BP measurement recorded in the preceding 5 years	295	280	95%
14.	Patient had a hypertension review with a doctor recorded in the last 12 months	295	49	17%
15.	Patient who has a hypertension review with a nurse/doctor recorded in the past 6 months after the BP is controlled, for patients with uncontrolled BP	293	71	24%
16.	Patient had a hypertension review with a nurse/doctor recorded in their medical record one month after being in step 7 of the algorithm of hypertension management, for patients already on medication	0	N/A	N/A
17.	Patient with a new diagnosis of hypertension aged 18–84 years, recorded (excluding those with pre-existing CHD, stroke and/or TIA), who had a recorded CVD risk assessment score of >20% in preceding 12 months	0	N/A	N/A
18.	Patients who are currently treated with statins (unless there is a contraindication) in those patients with a new diagnosis of hypertension aged 18–84 years, recorded (excluding those with pre-existing CHD, stroke and/or TIA), who had a recorded CVD risk assessment score >20% in the preceding 12 months	0	N/A	N/A
19.	Patient with hypertension aged 18 to 74 years in whom there was an annual assessment of physical activity in the preceding 15 months	233	1	0%
20.	Patient in the practice/unit/facility who had been counselled about the importance of smoking cessation in the last 12 months	295	115	39%
21.	Patient in the practice/unit/facility who has been counselled about the importance of maintaining ideal body weight, i.e., BMI < 25 kg/m^2^, in the last 12 months	295	97	33%
22.	Patient in the practice/unit/facility who has been counselled about the importance of salt restriction with increased potassium intake from fresh fruits and vegetables in the last 12 months	295	118	40%
23.	Patient in the practice/unit/facility who has been counselled about the importance of reducing alcohol intake to no more than 2 standard drinks per day for males and 1 for females in last 12 months	295	117	40%
24.	Patient in the practice/unit/facility who has been counselled about to follow a healthy eating plan in the last 12 months	295	119	40%
25.	Patient records with evidence that the nurse/doctor counselled the patient on the importance of engaging in physical activity, eating small portions of healthy food, using less salt, using alcohol in moderation, stopping smoking, reducing stress, committing to take medication regularly	295	97	33%
26.	Patient in the practice/unit/facility who has been counselled about the importance of engaging in regular moderate aerobic exercise, e.g., 40 min brisk walking at least 3 times a week, in the last 12 months	295	103	35%
27.	Patients diagnosed with hypertension who are given lifestyle advice in the preceding 12 months for smoking cessation, safe alcohol consumption, and healthy diet	138	17	12%
28.	Patient with hypertension and a BMI of ≥27.5 kg/m^2^ or ≥30 kg/m^2^ in the preceding 12 months referred to a weight management programme within 90 days of the BMI being recorded	75	0	0%
29.	Patient had cholesterol recorded in the last 12 months	295	35	12%
30.	Patients have heart/pulse recorded in the last 12 months	295	255	86%
31.	Patient had heart/pulse recorded in their medical record in the last 6 months	295	228	77%
32.	Patient had random blood glucose (≥11.1 mmol/L)/fasting blood glucose (≥7.0 mmol/L) recorded in past 6 months for all adults patients who are >40 years old and who are overweight (BMI > 25) or obese (BMI > 30)	97	7	7%
33.	Patient tested for the presence of protein in the urine by sending a urine sample for estimation of the albumin–creatinine ratio in the last 12 months	295	0	0%
34.	Patients with a new diagnosis of hypertension who have a record of a test for haematuria in the three months before or after the date of entry to the hypertension register	0	N/A	N/A
35.	Patients with a new diagnosis of hypertension who have a record of urinary albumin–creatinine ratio test in the three months before or after the date of entry to the hypertension register	0	N/A	N/A
36.	Patient with a BP of <140/90 mmHg with no adverse medicine reactions in patients who are in step 2 of the algorithm of hypertension management for patients already on medication every six months	108	62	57%
37.	Patient with a BP of <140/90 mmHg with no adverse medicine reactions in patients who are in step 3 of the algorithm of hypertension management for patients already on medication	124	38	31%
38.	Patient with a BP of <140/90 mmHg with no adverse medicine reactions in patients who are in step 4 of the algorithm of hypertension management for patients already on medication every six months	18	5	28%
39.	Patient with a BP of <140/90 mmHg with no adverse medicine reactions in patients who are in step 5 of the algorithm of hypertension management for patients already on medication every six months	6	3	50%
40.	Patient with a BP of <140/90 mmHg with no adverse medicine reactions in patients who are in step 6 of the algorithm of hypertension management for patients already on medication every six months	0	N/A	N/A
41.	Patient with a BP of <140/90 mmHg with no adverse medicine reactions in patients who are in step 7 of the algorithm of hypertension management for patients already on medication every six months	1	0	0%
42.	Patient referred to the doctor/district level services in the last 12 months	295	49	17%
43.	Patient had referral and reasons for referral in the last 12 months	49	46	94%
44.	Pregnant patients who were referred to district hospital services because they had severe pre-eclampsia and imminent eclampsia	0	N/A	N/A
45.	Patient received all core CVD/hypertension drugs	295	112	38%

NB: BMI, body mass index; BP, blood pressure; CHDs, coronary heart diseases; CVD, cardiovascular disease; eGFR, estimated glomerular filtration rate; TIA, transient ischemic attack. The indicators are sourced from the previous study [26].

## Data Availability

Data are available upon reasonable request.

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
