# Peer review of "Quality of Care for Hypertension in Primary Health Care in South Africa: Cross-Sectional Feasibility Study"

_healthcare, 2025, doi:10.3390/healthcare13192398_

Round 1

Reviewer 1 Report

Comments and Suggestions for Authors

This manuscript evaluates the quality of hypertension care in 12 South African primary healthcare clinics using 46 predefined indicators, revealing major gaps in adherence to clinical guidelines, data quality, and patient management. While the study highlights important systemic issues, it suffers from limited generalizability, unclear indicator applicability, and weak integration of contextual factors affecting care delivery.

  • The paper doesn’t explain how the 46 quality indicators were developed, which weakens their credibility.
  • The 75% threshold for indicator acceptability feels arbitrary and lacks justification.
  • The gender imbalance in the sample (77% female) is noted but not critically examined.
  • The study ignores patient perspectives entirely, reducing the richness of the findings.
  • Aggregating results across all 12 clinics masks potentially important variations.
  • Claims about clinimetric validity are made without showing how validity was actually assessed.
  • The discussion of weak health information systems is vague and lacks concrete details.
  • It’s unclear why some indicators couldn’t be applied—missing data and irrelevance are not clearly distinguished.
  • The reliance on paper records is acknowledged but not critically addressed in terms of its impact.
  • The discussion section is repetitive and lacks clear focus.
  • The self-citation rate is relatively high at 22% (13 out of 58 references), raising concerns about potential over-reliance on the authors' prior work.
  • Table 2 shows a high similarity index in the iThenticate report, apparently due to its close resemblance to content in the authors’ earlier publication (PMID: 39402282). This overlap should be addressed to ensure originality and transparency.
Comments on the Quality of English Language

While the overall meaning is understandable, there are recurring issues that affect clarity, flow, and professionalism. 

Reviewer 2 Report

Comments and Suggestions for Authors

The manuscript addresses an important and timely topic—quality of care for hypertension in primary healthcare settings—and the authors are to be commended for their efforts in exploring this area.

However, after a careful and thorough review, I regret to state that I am unable to recommend the manuscript for publication in its current form.

The present study suggests that evidence-based clinical guidelines are not consistently implemented, and treatment targets are currently not being achieved in primary healthcare (PHC) clinics in South Africa. The study should explicitly specify which clinical guidelines are recommended for adherence.

The methodology lacks sufficient depth and clarity to ensure reproducibility.

Since the indicators were extracted from medical records, it remains unclear how these can be applied directly to patients during in-person consultations. The study should also explore which additional indicators, particularly those related to lifestyle modifications can be assessed or incorporated beyond the existing medical record data to effectively manage and overcome hypertension.

Does "numerator met" indicate that the patient consistently adhered to the indicator throughout their follow-up treatment? Additionally, how was the treatment progress of patients who did not meet the numerator assessed or addressed?

What were the criteria used to determine adherence to the indicators? Additionally, for the indicators that were not achieved, what were the underlying reasons for their non-attainment?

Reviewer 3 Report

Comments and Suggestions for Authors
  1. In the title, authors should not leave more than one space between the words.
  2. In the Abstract, a sentence should be added noting the importance of quality care in patient care in hypertension specifically.
  3. I would suggest that authors present the objective (in the introduction) in a complete sentence and not use a, b, c.
  4. The structure of the manuscript and, subsequently, of the abstract should be Introduction (including the objective), Methods, Results, Discussion (including the limitations), and Conclusions. The Design and Setting should be included in the Methods section.
  5. In the study design, the authors should note that it is a retrospective study.
  6. European Society of Hypertension (ESH) recognizes and endorses the important role of Primary healthcare (PHC) in the diagnosis and management of hypertension. The ESH position paper by R Pihno et al. is suggested to be added as a reference to the introduction of the manuscript. The suggested manuscript is the following: Pinho R, et al. European Society of Hypertension - general practitioners' program hypertension management: focus on general practice. Blood Press. 2023 Dec;32(1):2265132. doi: 10.1080/08037051.2023.2265132. PMID: 37840300.
  7. Regarding the epidemiology of hypertension, authors should add references from large worldwide epidemiological studies. Moreover, in 2023, the WHO published reports on the prevalence and control of hypertension in most countries. Authors could add data from these reports to their manuscripts.
  8. Authors should ensure that references 9 to 12 include data from South Africa and data from other Southern African countries.
  9. Apart from South Africa, emerging evidence from European Countries (for Example, in Greece) suggests that in PHC physicians do not measure blood pressure (BP) in the majority of patients and their measurements do not comply at all with published guidelines. These data could be added to the discussion of the manuscript and discussed along with the findings of this study. The above data are included in the following abstract from ESH 2025: doi:10.1097/01.hjh.0001116508.57028.4e.
  10. The word blood pressure could be abbreviated as BP.
  11. I suggest that the authors present interquartile range (IQR) as: IQR (Q1- Q3).
  12. In Table 1, I suggest that variables should be presented according to their distribution. Variables with normal contribution should be presented as: Mean + Non-normal distributed variables should be presented as: Median (IQR), where IQR: Q1- Q3. Categorical variables are suggested to be presented as follows: % (n) or n (%).
  13. In Table 1, obesity should be defined as >30 kg/m2.
  14. Additionally, all the abbreviations from Table 1 should be explained as a footnote under the Table.
  15. Furthermore, was a reference to a specialized center for the evaluation of secondary hypertension (e.g., primary aldosteronism) incorporated into the quality assessment protocol?
  16. A significant limitation of the study is the retrospective design, something that should be underlined in the discussion section.

Round 2

Reviewer 1 Report

Comments and Suggestions for Authors

No comments

Author Response

No comments

Thank you for reviewing and providing comments and advice on our manuscript. We value your time and expertise. Thank you.

Reviewer 2 Report

Comments and Suggestions for Authors

The authors have addressed the majority of the concerns raised in the previous round, and the revisions have substantially improved the clarity and rigor of the manuscript. Minor language edits and formatting adjustments are still recommended to enhance readability, but no further substantive changes are required. I recommend acceptance after these small corrections.

Author Response

The authors have addressed the majority of the concerns raised in the previous round, and the revisions have substantially improved the clarity and rigour of the manuscript. Minor language edits and formatting adjustments are still recommended to enhance readability, but no further substantive changes are required. I recommend acceptance after these small corrections.

Thank you for your comment and advice. We have now edited the whole manuscript with the assistance of the two co-authors, who are native English speakers. We hope this is acceptable.

Thank you for reviewing and providing comments and advice on our manuscript. We value your time and expertise. Thank you.

Reviewer 3 Report

Comments and Suggestions for Authors

The authors should carefully check the references and the DOI added in every publication. 

Author Response

The English could be improved to more clearly express the research.

Thank you for pointing this out. We have now worked on language editing on the whole manuscript, with the assistance of the two co-authors who are native English speakers. We hope this is acceptable.

Reviewer comment: The authors should carefully check the references and the DOI added in every publication.

Thank you for your comment and advice. We have checked the references and added or amended the DOI in all the references as needed. All the amendments are highlighted in yellow. Thank you.

Thank you for reviewing and providing comments and advice on our manuscript. We value your time and expertise. Thank you.